# Impact of population pressure on forest resources depletion in Yayo coffee forest Biosphere Reserve, Southwest Ethiopia

Fikru Mosisa Hunde[1]*, Adanech Asfaw Benti[2], Tefera Jegora Kapula[2]

**1** Department of Biology, College of Natural Science, Mattu University, Mattu, Ethiopia, **2** Department of Forestry, College of Agriculture and Forestry, Mattu University, Mattu, Ethiopia

\* fikrumosisa@gmail.com

## Abstract

An increase in population density amplifies the demand for forest products, which in turn drives deforestation and the exhaustion of forest resource. The aim of the study was to assess the impact of population pressure on forest resource depletion in the Yayo Coffee Forest Biosphere Reserve, Southwest Ethiopia. A mixed-methods research design, integrating both quantitative and qualitative approaches. Data was collected from selected households, focus groups and key informant through semi-structured interview, group discussion and field observation. A systematic random sampling technique were used to collect the data. Data were analyzed quantitatively and qualitatively by SPSS and Microsoft office excels. Land-use and land-cover (LULC) changes over the past forty years were analyzed using satellite imagery to assess the impact of population growth on forest dynamics. Results indicated that the major livelihood strategies were contributed by the combination of crop production, livestock and forest product collection (36.2%) and followed by crop production and livestock (27.5%). Agricultural expansion (23.13%), overgrazing (17.9%), timber extraction (15.27%) and urbanization (14%) were the main direct drivers of forest loss. Satellite analysis revealed that forest cover declined from 120,087.2 hectares in 1982–100,772.9 hectares in 2024 an 11.6% reduction over four decades with a strong negative correlation (r = −0.998, p < 0.05) between population growth and forest area. Overall, both local practices and systemic pressures drive the conversion of forests to agricultural land in the Yayo Biosphere Reserve. This calls for district-specific interventions that engage indigenous institutions such as Shane, Xuxee, and Tuullaa in the management of coffee forests and enforcement of the law, encouraging alternative sources of energy, and ensuring that forest resources are not overexploited.

**Data availability statement:** The dataset supporting the findings of this study is publicly available in Dryad at https://doi.org/10.5061/dryad.66t1g1kfb. The data are fully anonymized and contain only the variables necessary to replicate the results reported in this study.

**Funding:** The author(s) received no specific funding for this work.

**Competing interests:** The authors have declared that no competing interests exist.

**Abbreviations:** CBOs: Community Based Organization; DAs: Developmental Agents; EARO: Ethiopian Agricultural Research Organization; FAO: Food and Agricultural Organization; IUCN: International Union for Conservation of Nature and Natural Resources; m.a.s.l.: Meter above sea level; MAB: Program on Man and the Biosphere; NGOs: Non-Governmental Organizations; SPSS: Statistical Package for Social Sciences; STCP: Sustainable Tree Crops Program;UNESCO: United Nations Educational, Scientific and Cultural Organization; ZEF: Zentrum fur Entwicklungsforschung (Center for Development Research).

# 1 Introduction

Ethiopia has undergone serious demographic growth in recent decades: for instance, the national population is estimated to be about 114.96 million in 2022, with an annual growth rate of about 2.6% [1]. This demographic expansion has increased pressure on natural resources, mainly forests, for agricultural land, settlement, grazing, and fuelwood purposes [2,3]. Population growth, together with rural livelihood dependency on biomass energy, has accelerated forest conversion and degradation throughout the country [4,5]. Ethiopia's forest cover was estimated at about 30–40% in the early 20th century, but it has declined dramatically to approximately 11–12% in recent decades [6,7]. Empirical studies have established the relationship between population growth and its corresponding percentage of agricultural expansion as the leading drivers of deforestation [8–10].

The Yayo Coffee Forest Biosphere Reserve (YCFBR), located in southwestern Ethiopia, is internationally recognized as one of the last remaining refuges of wild *Coffea arabica* and a vital center of Afromontane biodiversity [11,12]. Established under UNESCO's Man and the Biosphere Programme, the reserve plays a critical role in conserving genetic resources, supporting community livelihoods, and promoting sustainable development and research [13]. However, YCFBR is facing growing socioecological pressures [14].

Land-use change and encroachment of agriculture, including smallholder farms expanding into coffee plantations and settlements, have been marked by widespread forest degradation and frequent conflicts over resource use and land rights between the communities and management authorities [15]. Coal mining explorations in the buffer zones, estimated at about 230 million tons of coal, have posed additional threats to forest integrity and Low productivity of smallholder coffee, ranging between 400 and 600 kg ha$^{-1}$, against potential yields exceeding 1,000 kg ha$^{-1}$, is also among the key drivers of conversion to expand cultivation areas [16].

The associated biodiversity and ecosystem service decline are evident, with highly simplified monocropping systems supporting significantly lower species diversity compared to more complex agroforestry mixtures [17]. Yayo Coffee Forest Biosphere Reserve is continuously experiencing increased deforestation, agricultural expansion, and socio-economic tensions [18]. Rapid population growth, accompanied by a decline in farm productivity due to land degradation and soil erosion, along with very limited livelihood diversification strategies, has increased local dependence on forest resources including timber, charcoal, and fuelwood [19].

Addressing these gaps is critical because YCFBR preserves the wild gene pool of *Coffea arabica,* vital for global coffee sustainability [20]. Understanding livelihood and forest interactions can offer guidance to how balanced management can provide basic needs to the local community, sustain biodiversity, and contribute to the wider debates on tropical forest governance and ecosystem-based livelihoods [21]. The study, therefore, intends to identify the effects caused by population pressure on forest resource depletions in the Yayo Coffee Forest Biosphere Reserve, Ilubabor Zone, Oromia Regional State, Southwest Ethiopia.

## 2 Methods and materials

### 2.1 Study area description

**2.1.1. Location.** The Yayo Coffee Forest Biosphere Reserve is located in the Ilu Abba Bora Zone of the Oromia Regional State in southwestern Ethiopia. It is recognized as the center of origin for *Coffea arabica*, the world's most widely consumed coffee, and Yayo is the largest and most significant forest for conserving wild coffee populations globally [22]. The reserve plays a crucial role in preserving both natural and cultural landscapes [23]. Situated in the southwestern part of the Oromia region, the Yayo Biosphere Reserve includes the Woredas of Hurumu, Yayo, Chora, Nopha, Alge Sachi, and Doreni, covering coordinates from 8°0'42" to 8°44'23" N and 35°20'31" to 36°18'20" E [Fig. 1]. The district's elevation

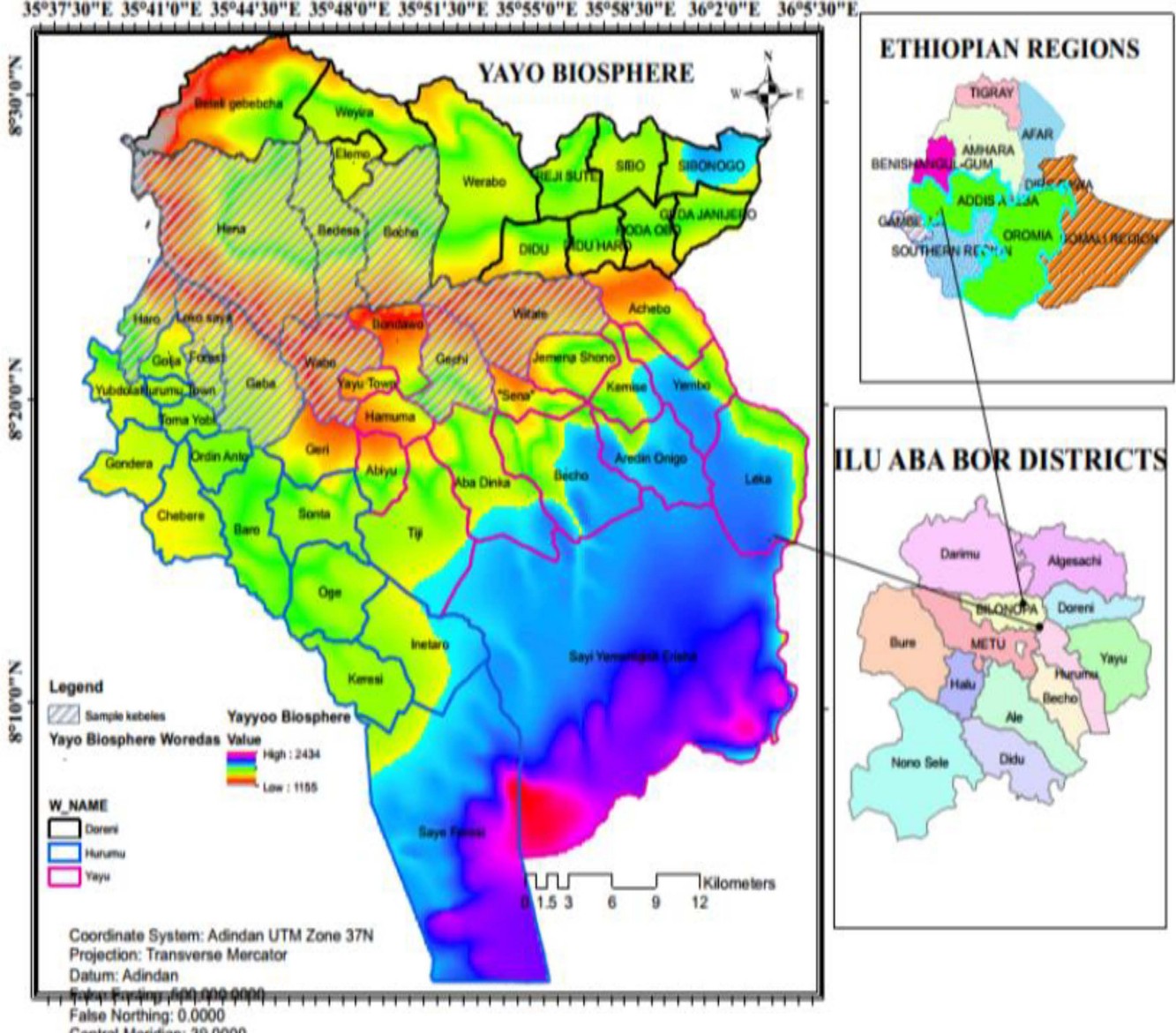

**Fig 1. Maps of the study area [Yayo coffee forest Biosphere Reserve, Southwest Ethiopia, 2024].**

ranges from 1,139.2 to 2,581.9 meters above sea level, with the lowest point located at Gaba River (1,100 meters) and the highest point at the summit of Sayi Mountain (2,581 meters) in Keresi.

**2.1.2. Climate.** The area experiences a hot and humid climate, with an average annual temperature of around 23°C, ranging from a mean minimum of 18.59 °C to a mean maximum of 27.88 °C [24]. The varied physical conditions and altitudinal differences contribute to a rich diversity of climate, soil, and vegetation, fostering the development of numerous plant species with a high level of diversity [25]. The rainfall pattern of the districts varies annually from 1,191.6 to 1,960.7 mm, showing variations from year to year. It is a unimodal type of rainfall that increases from May to October and declines in November [26].

**2.1.3. Ecology.** The study focuses on the Hurumu, Yayo, and Doreni districts located in the Yayo Coffee Forest Biosphere Reserve in southwestern Ethiopia, which is famous for its ecological importance and genetic diversity of wild *Coffea arabica* [27]. The biosphere reserve has a total surface area of 167,021 hectares and consists of three zones: a core zone (27,733 ha), a buffer zone (21,552 ha), and a transition zone (117,736 ha), balancing conservation and sustainable development [28]. Land in the study area comprises 3.5% highland (5,750.4 ha), 85% temperate zone (138,465.85 ha), and 11.47% lowland (18,684.75 ha), reflecting its heterogeneity and altitudinal gradients [29]. This diversity accounts for much of the biodiversity of the region and explains its status as a key location for environmental studies and resource management [30].

The diverse climatic conditions and habitats in these districts have contributed to a high level of species diversity in both plants and animals [31]. This biodiversity richness is one of the reasons why Ethiopia is considered one of the 20 most biodiverse countries in the world [32].

## 2.2 Study design

This researcher employs a mixed-methods research design, integrating both quantitative and qualitative approaches to comprehensively investigate the impacts of population dynamics on the Yayo Biosphere Reserve. The design is structured to systematically address the following objectives: assessing population growth, evaluating forest cover change, examining conservation practices, understanding community resource relationships, identifying stakeholder roles, and proposing sustainable management strategies.

## 2.3 Data collection tools

Following ethical approval, the study was conducted in several data collection waves between May 2024 and January 2025. The data for this study includes a sample household survey and land use/land cover change data. The research approach involves a consultative and interactive process, engaging respondents who are willing to provide essential information [33].

**2.3.1. Key informant interviews (KIIs).** Key informant interviews were conducted with individuals who possessed in-depth knowledge and experience related to forest resource management and population dynamics in the area. These informants included community elders, local administrators, development agents, forestry experts. Interviews were conducted to collect data on the role of local communities in forest conservation, the impacts of human activities on the forest, and perceptions toward conservation efforts, as well as the challenges and consequences associated with these activities [34].

**2.3.2. Household surveys.** Household surveys were used to collect quantitative data from a representative sample of households within the Yayo Coffee Forest Biosphere Reserve. Structured questionnaires were administered to gather information on demographic characteristics, household size, landholding, livelihood activities, fuelwood consumption, and perceptions regarding forest resource use and conservation. This tool enabled statistical analysis and comparison across different groups and areas [35].

**2.3.3. Focus group discussions (FGDs).** Focus group discussions and interviews were conducted to collect primarily qualitative data. Satellite imagery was utilized to generate land use land cover change data [36]. These informants

included village elders, kebele administration, development agents (DAs). Three FGDs were also organized to garner community-level perspectives regarding forest management and socio-economic issues [37]. These interviews enriched the qualitative data through the locally grounded knowledge and lived experience [38].

**2.3.4. Observations.** Field observations were carried out to supplement data obtained through surveys and interviews. These observations focused on land-use patterns, forest cover conditions, agricultural expansion, and evidence of forest degradation. Observation checklists were used to ensure systematic recording of data. This method helped verify and validate information gathered from other sources and provided a practical understanding of the extent of forest resource depletion in the study area.

## 2.4 Target populations

The population of interest for this study comprised rural households and key stakeholders residing in and around the Yayo Coffee Forest Biosphere Reserve, specifically within the selected woredas of Yayo, Hurumu, and Doreni. During the study time, there were a total of 91,694 households; 89,039 were males and 2,659 were females, with a total population of 458,472 individuals. These communities were chosen due to their direct dependence on forest resources for their livelihoods and their proximity to the core and buffer zones of the biosphere reserve [39,40].

The sampling frame included residents from nine kebeles, three from each of the selected woredas, identified based on geographic location, accessibility, and relevance to forest conservation activities [41]. Households were selected using purposive sampling techniques to ensure the inclusion of individuals with lived experience in forest resource use and conservation [42]. Criteria such as length of residence in the area, proximity to the forest, and knowledge of local forest management practices were considered during the selection process [43].

## 2.5 Sampling methods and techniques

To ensure a comprehensive understanding of the study area, a combination of preliminary assessments and quantitative and qualitative surveys was employed [44]. An initial exploratory survey was conducted to gain an overview of the spatial distribution of the Yayo Coffee Forest Biosphere Reserve and to facilitate the selection of appropriate study sites [27]. From a total of 91,694 households residing in the study woredas comprising 89,039 male-headed and 2,659 female-headed households a sample of 156 households was selected for the survey. A systematic random sampling technique was employed to ensure that the selected sample was both representative and unbiased.

The Yayo Coffee Forest Biosphere Reserve, which spans six woredas of the Ilubabor and Buno Bedele zones namely Yayo, Hurumu, Chora, Bilo Nopha, Alge Sachi, and Doreni served as the general study area. For this particular study, Yayo, Hurumu, and Doreni woredas were selected as representative sites. Within these woredas, nine rural kebeles were purposively chosen based on their proximity to the forest zone and relevance to forest resource use: Haro, Gaba, and Wangegne from Hurumu woreda; Waboo, Geci, and Wixete from Yayo woreda; and Bocho, Badesa, and Henna from Doreni woreda. These kebeles were chosen on purposively since they are nearest to forest, residence in the biosphere reserve for years, and knowledge on forest resource use and conservation methods [45].

To gather quantitative information, a systematic household survey was conducted with 156 respondents from the chosen kebeles. The survey collected baseline data on household characteristics, land use, livelihood strategies, and perceptions towards forest conservation [46,47]. At the same time, qualitative information was gathered through key informant interviews and FGDs. 18 key informants, two from each kebele, were interviewed Three FGDs were made, each containing six members, one from each woreda, resulting in a total of 18 participants.

## 2.6 Data analysis

The quantitative data collected through household surveys were first cleaned, validated, and coded in Microsoft Excel to promote consistency and accuracy [48]. The cleaned data were then exported into IBM SPSS version 20 for analysis. Besides, tables, graphs and charts were implemented using descriptive statistics; frequencies, percentages and other statistical analyses. Descriptive statistics were used in summarizing key variables such as household characteristics, forest resource use, and conservation knowledge [49].

To analyze causal relationships linear regression and logistic regression tests were used. These methods allowed the assessment of the impact of independent variables, i.e., household size, income level, and education, on dependent variables like dependency on forest resources and conservation behavior [50]. The validity of the regression models was checked by using diagnostic tests [51].

For the qualitative data collected through key informant interviews and focus group discussions, thematic analysis was applied [52]. The data were coded, transcribed, and categorized into local perceptions, emerging themes, traditional knowledge, forest governance, and livelihood strategies. Representative statements of the participants were utilized to elaborate on these themes [53]. Through this mixed-methods approach with regression-based quantitative analysis complemented by rich qualitative feedback, there was a guaranteed holistic understanding of the drivers, effects, and community perspectives about the use and preservation of forest resources [54].

## 2.7. Ethical considerations

Ethical approval was granted by the Mattu University Institutional Review Board (IRB) prior to participant recruitment and data collection. The ethical clearance was issued under Reference Number 258/MaU/2024 on May 12, 2024. The purpose and procedures of the interviews were clearly explained to all study participants before obtaining their consent to participate. Verbal informed consent was obtained from all stakeholders, each of whom retained a copy of the consent form for reference. Participation was entirely voluntary, and participants were free to withdraw at any time without penalty. All collected data were securely handled and stored to ensure participant privacy, with access restricted solely to the research team.

## 3 Results and discussion

### 3.1 Socio-economic characteristics

The majority of the households, 89,039 (97.1%), were male-headed, while only 2,659 (2.9%) were female-headed. The ages of the respondents ranged from 20 to 80 years, with a mean age of 48 years. The family size varied between 1 and 12 members, with an average of 5.18, which is consistent with the national average family size. Regarding education, 15.9% of the respondents were illiterate (unable to read or write), 30.4% had attended grades 1–5, 42% had completed grades 6–12, and 11.6% had completed secondary. On income, 14.2% of the households did not have any regular earnings or financial inflow (they were dependent on others), 39.8% of the households were low-income or below median household income (<10,000–15,000 ETB/year), 35.1% middle-income or, around median (15,000–50,000 ETB/year) and only 10.8% high-income or well above median (>50,000 ETB/year) [Central Statistical Agency of Ethiopia (CSA), Household Income and Expenditure Survey, 2015/16]. In terms of household size, the largest share (55.4%) included 4–5 members, then 23.6% included 6–7 members, and 12.8% included over 7 [Table 1]. Similarly, the family's socioeconomic status includes the household income, earners' education and occupation, as well as combined income when their own attributes are assessed [55]. In fact, the socio-economic status can be measured in a number of different ways, and most commonly, it is measured by education, occupation, and income [56]. Family illiteracy drastically restricts economic opportunities via limited availability of well-paying employment, resulting in poverty; affects the health services; and is unlikely to possess the capacity to aid their education [57,58].

**Table 1. Socio-demographic and economic characteristics of the participants among households in Yayo coffee forest Biosphere Reserve, Southwest Ethiopia, 2024.**

| Variables | Category | Frequency | Percent (%) |
|---|---|---|---|
| Sex | Male | 144 | 97.2 |
| | Female | 4 | 2.8 |
| | Total | 148 | 100.0 |
| Age | 20-30 | 4 | 2.9 |
| | 31-40 | 58 | 39.1 |
| | 41-50 | 34 | 23.2 |
| | 51-60 | 22 | 14.5 |
| | >61 | 30 | 20.3 |
| | Total | 148 | 100.0 |
| Educational status | Illiterate (no formal education) | 24 | 15.9 |
| | Grade 1–5 | 45 | 30.4 |
| | Grade 6–12 | 62 | 42.0 |
| | Diploma & above | 17 | 11.6 |
| | Total | 148 | 100.0 |
| Income level | No income | 21 | 14.2 |
| | Low income | 59 | 39.8 |
| | Middle income | 52 | 35.1 |
| | High income | 16 | 10.8 |
| | Total | 148 | 100 |
| Family size | ≤ 3 | 12 | 8.0 |
| | 4–5 | 82 | 55.4 |
| | 6–7 | 35 | 23.6 |
| | > 7 | 19 | 12.8 |
| | Total | 148 | 100.0 |

## 3.2. Livelihood strategies

In this study, among the sample households that participated in the survey, 36.2% of respondents indicated that their income came from crop, livestock and forest product, 27.5% from crop production and livestock rearing, and 15.2% from forest product collection and off farm activities. Additionally, 10.2% reported that their income sources were from crop production only. 6.9% of respondents indicated that their income relied on livelihood activities and 4% depends on off farm activities [Fig 2]. Against the above result, 73% of forest dwellers depend on these products as a source of revenue in the Bale Zone, southern Ethiopia [59]. The value of income sources like forest products, crop production, livestock production, and off-farm activities depends on each other [60]. Research conducted in the adjacent district of Yayo, included under one National Forest Priority Area (NFPA) with Gabba-Dogi, i.e., Yayo NFPA [61], revealed that 92.6 percent of the population in the study area have coffee in the forest, from which 57.3 kg of honey on average is harvested per household per year [62].

## 3.3. Formal institutions in forest management

Based on the information gathered from interviews and focus group discussions, various institutions at the federal, regional, and local levels were identified, along with their interlinkages concerning the management of the coffee forest resource under investigation [63].

Changes in the institutional structure of the Ministry of Agriculture (MoA) since the early 1990s have failed to establish a dedicated government body for the management, conservation, and sustainable use of coffee forests [22]. This gap

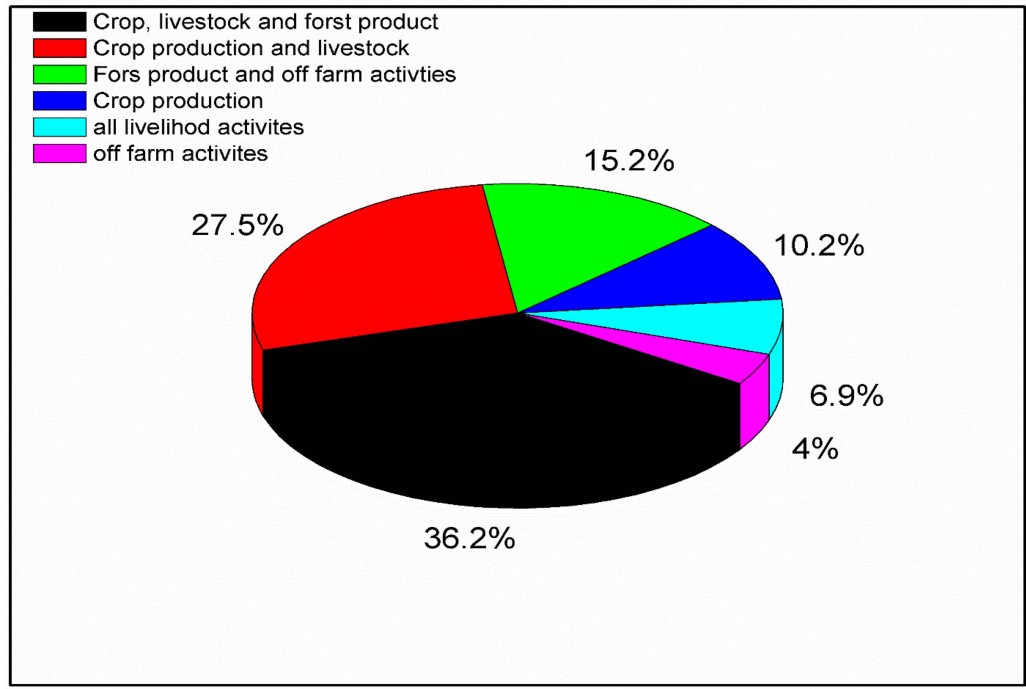

**Fig. 2. Livelihood strategies among study participants [Yayo coffee forest Biosphere Reserve, Southwest Ethiopia, 2024].**

has contributed, to some extent, to the increasing lack of effective institutions for the sustainable management of coffee forests. Federal institutions, mainly under the Institution for Biodiversity Conservation (IBC), provide technical support for Yayo Coffee Forest Biosphere Reserve conservation [64].

At the regional level, two institutions with potential connections to the coffee forest are examined: the Oromia Forest and Wildlife Enterprise Supervising Agency and the regional agriculture and rural development bureaus established in 1999 under proclamation 90/1999 as parts of decentralization [55]. The primary issues identified within these organizations include a lack of technical and direct focus on forest coffee biodiversity conservation, insufficient budget and technical personnel at the agricultural and rural development bureaus, inadequate decentralization of the budget, and limited community involvement in planning and execution at the Forest Enterprise Supervising Agency [65]. Another gap noted in the State Forest Enterprise is the absence of incentives to motivate local communities to conserve specific forests [66]. The study confirms that local indigenous institutions have been gradually marginalized due to the increasing control of the state, which enforced formal institutions under various regimes in the past [67].

At the local level, the institutions involved in the use, conservation, and management of the coffee forest include the Yayo Coffee Forest Biosphere Reserve Conservation Project, the district administration (comprising the kebele and development team), and the district [68].

### 3.4. Informal institutions contributing for forest management

The study examines various informal institutions in terms of their structure and role in the livelihoods of the community in the study area. These institutions are categorized into four groups, with particular emphasis placed on two clusters [69,70]. The territorial-based administrative indigenous/customary institutions are further divided into four groups: Tuullaa, Xuxee, Shane, and Jaarsa Biyya, along with Muchoo. The study conducted by Tulu and Getahun revealed the same result

with current study. The first cluster consists of territorial-based administrative indigenous/customary institutions, while the second includes a variety of self-help work organizations [70].

Shane, Xuxee, and Tuullaa make two important contributions to the management of coffee forests. A fundamental role in bringing local governance systems into alignment is played by Tuullaa, which organizes, leads, and enforces the work, norms, and regulations of various local customary institutions and self-help labor associations. By controlling coffee harvesting and enforcing customary regulations to stop overexploitation and resource degradation, for example, these organizations manage coffee forests both directly and indirectly [69]. This finding aligns with the broader literature on common-pool resource management, which emphasizes that strong social norms, particularly those founded on reciprocity and generalized trust, are instrumental for the sustainable management and conservation of collective natural resources. The result highlights the central role of Tuullaa as a culturally grounded institution that governs not only the social and economic existence of the community in the coffee forest zone but also exercises a significant influence over governing the environment [71].

The ability to govern conduct among community members and facilitate the crafting and enforcement of local rules situates it as a significant mechanism for maintaining collective action for sustainable forest use. This is in harmony with common-pool resource governance theory, which praises the effectiveness of locally crafted institutions in shared ecosystem management [71,72].

Jaarsa Biyya and Muchoo, as customary Oromo institutions, play a crucial role in the sustainable management of the coffee forest and other natural resources by being local enforcers of customary law and mediators in resource disputes. Their authority, which has its foundation in respect by the community, allows them to enforce existing rules and, if supported by the locals, create, and implement new rules. This aligns with the contention that locally embedded institutions are more effective and legitimate in the governance of resources than externally introduced formal institutions [72]. Similarly, it contends for the importance of facilitating resource users to participate in rule-making, a function these institutions are strategically positioned to undertake [73]. This is also supported by who adds that local self-help associations among the Oromo have traditionally worked to improve food security and economic resilience [74,75]. Interpreted through the theory of institutional bricolage, Jaarsa Biyya and Muchoo demonstrate how traditional authority can be reshaped to contemporary challenges, blending cultural practice with evolving governance imperatives for promoting sustainability [76].

### 3.5. Conservation policies and its drawbacks

As indicated by various informants in the coffee forest study area, different forms of ownership rights existed prior to the demarcation process. The publicly owned and strictly conserved portion of 50,000 hectares of the Yayo Coffee Forest Biosphere Reserve is the most significant. The remaining 117,000 hectares fall under a mix of community, cooperative, and private regimes, which work to strengthen traditional forest coffee production. Such sites support over 150,000 people, with 40–50% of the reserve in mixed regimes [77]. Federal Forest Development, Conservation, and Utilization Proclamation no. (5, 6) has a sure implication for natural resource management, even though in the practical implementation they have faced their own limitation [78].

### 3.6. The contributions of the biosphere reserve to the local communities

In this study, the collection of forest products was identified as the primary source of household income. Approximately 43.9% of respondents stated that they are highly dependent on forest products to support their households. Additionally, 37.8% of the households surveyed indicated that collecting forest products is their top priority for sustaining their livelihoods, While 16.2% of the respondents said forest product collections are serving my household as a second income source, and 12.8% of them have ranked forest products as the third option for their livelihood [Table 2]. This finding is similar with many studies conducted in different areas in Ethiopia [79]. For instance, in the Yayo district, the same study area, in the Sheka zone, and in the Bale Mountains, forest products were revealed as the primary source of income, with

**Table 2. Forest products priority to support the household's livelihood among study participants in Yayo coffee forest Biosphere Reserve, Southwest Ethiopia, 2024.**

| Preference category | No. of respondents | Percentage (%) | Rank |
|---|---|---|---|
| General dependency on forest products | 65 | 43.9% | 1 |
| Forest products as 1st priority | 36 | 37.8% | 2 |
| Forest products as 2nd priority | 24 | 16.2% | 3 |
| Forest products as 3rd priority | 19 | 12.8% | 4 |
| Not engaged in forest products | 4 | 2.7% | 5 |
| Total | 148 | 100.0% | |

contributions of 54%, 49%, and 44.7%, followed by crop production [80]. In contrast, other studies have indicated that forest products contribute as the fourth most important source of livelihood for households. For instance, at Liban Woreda, Borena, southern Ethiopia (32%), and at Gore District, southwest Ethiopia, similar agro-ecology (23%) [81].

### 3.7. The major factors of forest depletion

The analysis of the household survey results highlighted several factors contributing to the depletion of forest stocks in the Yayo Coffee Forest Biosphere Reserve. Among the households surveyed, approximately 23.13% of respondents identified agricultural expansion as a major factor in the depletion of forest resources due to the local population's reliance on primary activities such as wood logging, non-timber forest product collection, and farming. The findings in the Yayo Coffee Forest Biosphere Reserve from the survey identify three major causes of forest degradation: lack of indigenous community participation (21.95%), Timber extraction (15.27%) and Urbanization (14%). The Yayo Fertilizer Factory, which has removed about 115 hectares of forestland from the buffer zone of Yayo Biosphere Reserve, is a poor example of industrial investment, which 12.2% of respondents cited as one of the primary causes of forest degradation. This is consistent with research showing that industrial operations significantly speed up deforestation and pose serious risks to biodiversity [Fig 3]. These are strongly supported by findings in some well-accredited studies in comparable environmental and socio-economic contexts. For instance, it is projected that urban expansion would account for the conversion of over 2.5 million hectares of forests globally by 2030, primarily in developing nations, as a testament to the impact of urban spread on forest degradation [82].

The respondents' high agreement with the statement that "the lack of indigenous involvement results in forest depletion" indicates that 73.9% of respondents strongly agree with this statement, which has been confirmed [83]. Similarly, the 75.7% who identified timber extraction as a major cause aligns with previous studies and reports, which rank unsustainable and illegal logging among the leading drivers of forest loss [84,85].

These trends emphasize the pressing need for more stringent environmental regulation and more environmentally friendly development practices. Moreover, the residents reported that the Yayo Fertilizer Factory displaced a high number of farmers from their farms, and they resorted to daily work as a source of survival [84,86]. The research revealed that approximately 2,200 farming families were physically displaced, losing their farms, houses, and wild coffee farms as sources of livelihood to accommodate the factory's production [87].

Additionally, the crew saw firsthand how more than 26 low-income families depended on daily wage labor to make ends meet. The Community Forest Coffee Project's (COFCOP) arrival was considered noteworthy in this context because it provided some alternative livelihood support [88].

The coal mining initiative, launched in 2010, aimed to generate employment for thousands of jobless locals in the region. Between 2012 and 2013, over 6,000 skilled and unskilled workers were hired by the Yayo Fertilizer Factory, with most resettled in heavily forested villages such as Achebo and Wutete. However, starting in 2018, coal mining at the Achebo site came under the control of organized youth from displaced families. As a result of the project's strict security measures, hundreds of villagers were physically displaced and denied access to their coffee farms, forest lands, and

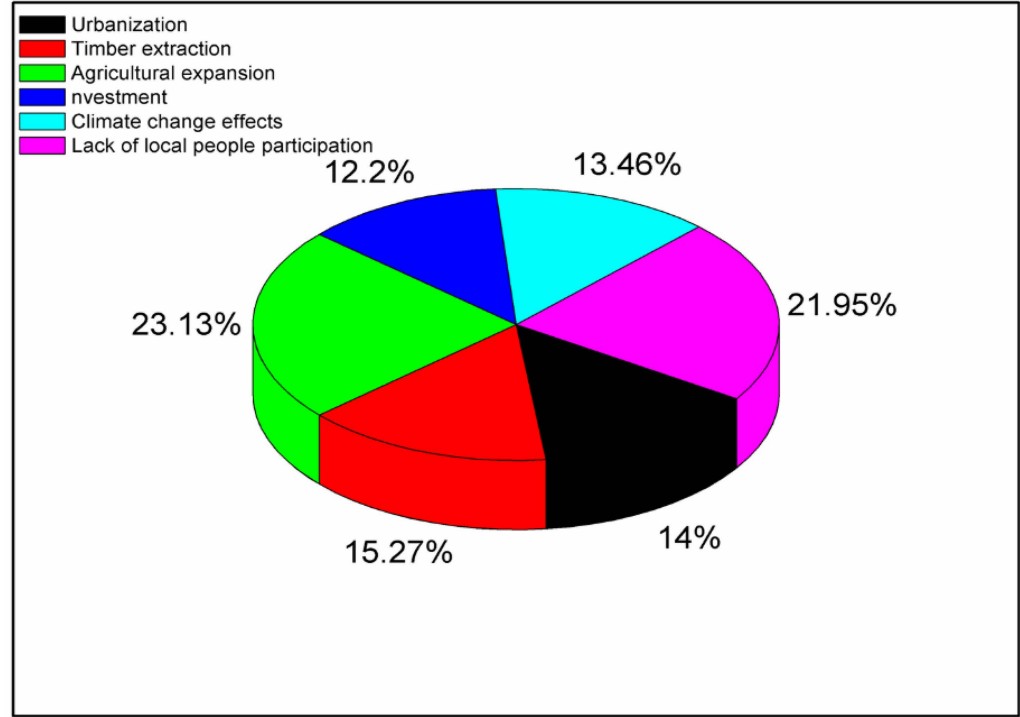

**Fig 3. Factors contributing for the possible depletion of forest stocks [Yayo coffee forest Biosphere Reserve, Southwest Ethiopia, 2024].**

water sources. While compensation was provided, it was limited to losses of seasonal crops (like maize, sorghum, and barley), perennial crops (such as mango, pawpaw, and oranges), garden trees, and village homes and structures. The study also revealed that compensation was calculated for only 420 of the 2,200 displaced individuals. For example, compensation for a coffee stand was estimated at just 6.50 birr, and for one hectare of farmland, it was only 400 birr [88].

Other study results indicated a lack of economic opportunities and social amenities in rural areas; rural-urban migration has resulted in a considerable population increase in many sub-Saharan African cities [89].

### 3.8. Population growth and forest cover change

According to the statistical findings from the household survey, the main drivers of this activity were identified as farmland expansion (40.5%), forest product collection (29.6%), settlement (12%), and overgrazing (17.9%), with these factors emerging as the most significant contributors. Ethiopia continues to have one of the highest population growth rates in sub-Saharan Africa, 2.5% to 3% annually, which adds pressure to land and other natural resources [90]. Over the past few decades, the region has been a big hub for migrants, with both rising numbers and rising ethnic diversity. Other groups who settled in the area include people from the Southern Nations, Nationalities, and Peoples' Region (SNNPR), Amhara, Tigray, and eastern Oromia [91]. Such movements relate to broader national patterns of domestic migration, often predicated on voluntary migration as well as government-sponsored resettlement programs aimed toward relatively sparsely populated forest zones [92]. Household survey reports that forest product collection, mainly firewood and forest coffee, as a leading cause of forest degradation. Hurumu Woreda informants report widespread overexploitation of tree species like *Cordia africana* for illegal timber [Fig 4]. This aligns with general trends in Ethiopia and sub-Saharan Africa, where wood fuel and timber overextraction drive deforestation [93]. *Cordia africana*, of economic value for timber and shade, is most susceptible, threatening biodiversity and agroforestry. Despite its importance for livelihood, forest product collection accelerates degradation, particularly when combined with agricultural encroachment and settlement [90].

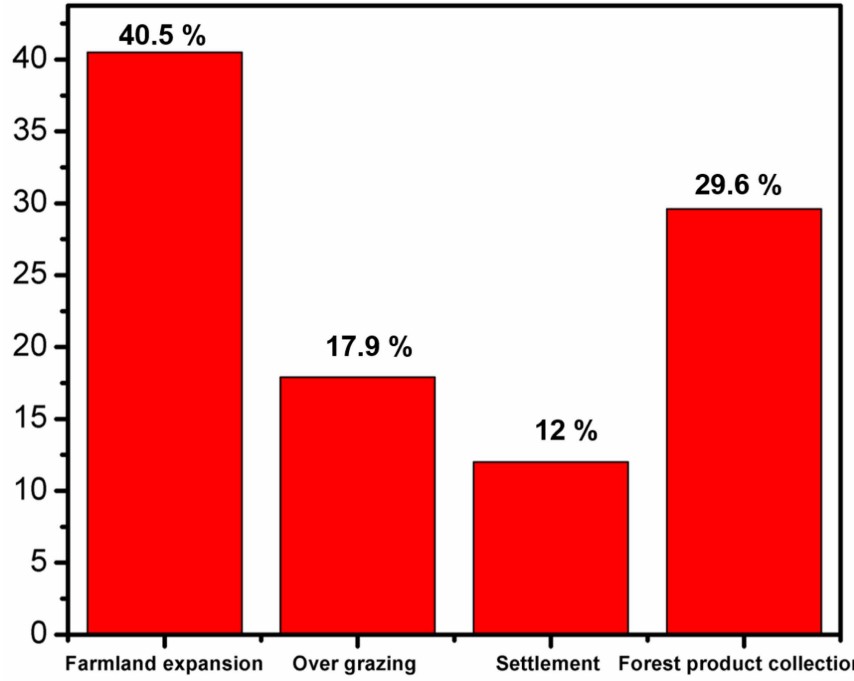

**Fig 4. Forest land Conversion [Yayo coffee forest Biosphere Reserve, Southwest Ethiopia, 2024].**

Farmland expansion, identified by 35.8% of respondents, is a leading cause of forest degradation in the area, while settlement activities contribute to 18.9% of the degradation [Table 3]. Overgrazing, however, is not as prominent since it entails the use of certain grazing fields (kaloo) that are different from forests. Similar findings in the Ethiopian highlands show that spatial land-use management, like exclosures, can avert degradation [94,95]. In contrast, regions lacking these management systems, such as Afar and Borana, are facing severe soil degradation and loss of biodiversity, largely driven by uncontrolled overgrazing and the absence of sustainable land-use practices [96,97].

### 3.9. Demographic trends from 1984 to 2024 and forest cover change satellite image result

Population data for the past 40 years were obtained from the Central Statistical Agency (CSA), Jimma Branch, based on the Ethiopian Population Censuses conducted in 1984 and 1994, as well as projections from the 2007 census. According to the census data, the population of Ilubabor Zone was over 847,048 in 1984, over 970,243 in 1994, 1,271,609 in 2007 and 2,301,242 in 2024. The population of Yayo district increased from 29,837 in 1984–52,851 in 1994, then to 83,579 in 2007, and further to 139,001 by 2024. Similarly, the population of Hurumu district was 27,586 in 1984, rising to 42,667 in 1994, 106,294 in 2007, and 165,143 by 2024. In Dorani district, the population was 20,015 in 1984, increased to 36,705 in 1994, reached 58,938 in 2007, and grew to 97,491 by 2024 [Fig 5]. The forest cover change in Yayo Biosphere Reserve over the past 40 years (from 1984 to 2024) is presented as a satellite image comparison. Landsat 5 TM for 1984, Landsat 5 TM and ETM for 1994, and Landsat 7 TM and ETM+ for 2024, each with a 30-meter resolution satellite image cover, showed Forest coverage decreased from 120087.2 hectares to 100772.9 hectares, or by 11.6%, over the 40-year period [Table 3]. Forest coverage is replaced by agricultural land, and settlement land is the dominant land use type in the study area, which is divided into three compartments: core, buffer, and transitional zones, within the Yayo Biosphere Reserve, and LANDSAT/TM satellite images from 1986 to 1990 show that Ethiopia's forest cover had since then been reduced to 3.93%, or 45,055 sq km [98].

**Table 3. Satellite images results of land use land cover in the Yayo coffee forest Biosphere reserve from 1984 to 2024 (S1 Fig).**

| Districts | Land use land cover | 1984 | | 2004 | | 2024 | |
|---|---|---|---|---|---|---|---|
| | | Area (Ha) | % | Area (Ha) | % | Area (Ha) | % |
| Doreni | Forest | 29030.1 | 63.1 | 25395.59 | 55.2 | 24889.52 | 54.1 |
| | Agriculture & Settlement | 16976.4 | 36.9 | 20610.91 | 44.8 | 21116.98 | 45.9 |
| | Total | 46006.5 | 100 | 46006.5 | 100 | 46006.5 | 100 |
| Hurumu | Forest | 30848.4 | 66.3 | 29638.65 | 63.7 | 24241.35 | 52.1 |
| | Agriculture & Settlement | 15680.1 | 33.7 | 16889.85 | 36.3 | 22287.15 | 47.9 |
| | Total | 46528.5 | 100 | 46528.5 | 100 | 46528.5 | 100 |
| Yayo | Forest | 60208.67 | 74.5 | 57703.34 | 71.4 | 51642.06 | 63.9 |
| | Agriculture & Settlement | 20608.34 | 25.5 | 23113.66 | 28.6 | 29174.94 | 36.1 |
| | Total | 80817 | 100 | 80817 | 100 | 80817 | 100 |
| Grand total Forest land | | 120087.17 | 69.3 | 112737.58 | 65 | 100772.93 | 58.1 |

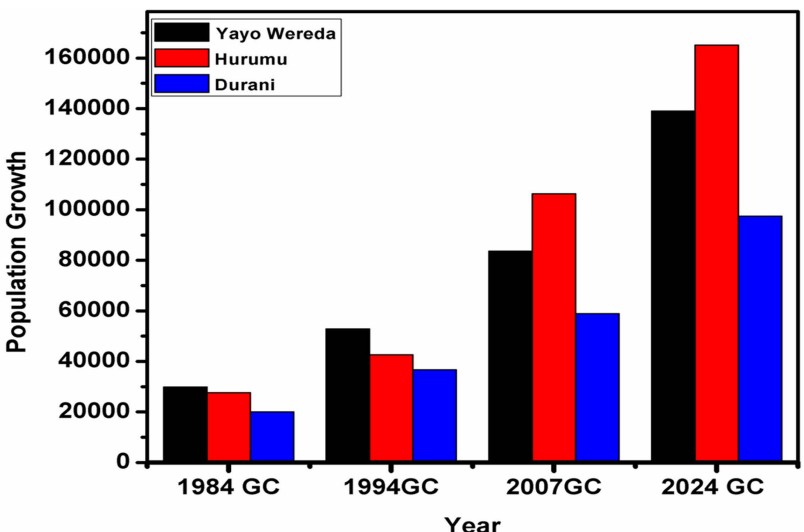

**Fig 5. Demographic Changes in the Yayo coffee forest Biosphere Reserve from 1984 to 2024, Southwest Ethiopia [Yayo, Hurumu, and Doreni Districts].**

The Yayo Coffee Forest Biosphere Reserve is a significant ecological location that sequesters carbon and mitigates climate change, which makes it both a national and global priority [99]. Agroforestry systems in the region, particularly those for coffee grown under shade, are a sustainable land use practice that conserves the environment and promotes local development and illustration of a climate-resilient approach in that they enhance carbon sequestration and sustain rural livelihoods [100].

The identification of these land use changes as significant contributors to the region's forest degradation further supported the findings of statistical analysis and satellite imaging, and it makes it exceedingly dangerous for the biosphere reserve to maintain the biodiversity that made it possible for it to be identified and About 15 households have unlawfully established on 8 hectares of the biosphere reserve's core zone, according to data acquired from focus groups and key informant interviews with farmers in Yayo District, particularly in the Kori area close to Sayi Forest [105]. Similarly, in Ilu Abba Dinka Kebele, Janeh area, about 28 households have settled on 14 hectares of the core zone. In total, approximately 22 hectares of forest in the core zone have been cleared due to agricultural expansion and coffee production. Additionally, in some areas, particularly in

Gechi Kebele, local people have raised concerns about the planting of exotic tree species, which they believe could rapidly threaten and negatively affect the indigenous species by clearing the native vegetation [101].

The present results align with a study conducted in the Hawa Galan district of Kelem Wollega, Ethiopia, which found that forest cover in the district was negatively correlated with both the district's overall population and the population within the forest area [102]. Another study reported that population growth has intensified pressure on scarce natural resources, leading to the degradation and destruction of forests and woodlands [103].

### 3.10. Socio-economic determinants of forest dependency and conservation behavior

The Pearson Correlation analysis results indicate a strong negative correlation between forest cover and population growth rate over the past 40 years in the surveyed forest biosphere reserve, with a coefficient value of −0.998. This negative value confirms an inverse relationship between population growth and forest cover, meaning that as population growth increases, forest area decreases, and vice versa [104]. The statistical analysis also shows that this correlation is significant at the 0.05 confidence level ($P < 0.05$) [Table 4]. In line with the above findings, the impact and pressure of population on land use and land cover (LULC) changes are strongly influenced by population density. Pearson correlation analysis showed that forest cover ($P = −0.006$) is negatively associated with population pressure on forests [104].

Based on the Data from 148 households in the Yayo Coffee Forest Biosphere Reserve identified key socio-economic factors influencing forest dependence. Multiple linear regression analysis showed that household size ($\beta = −0.32$), income ($\beta = −0.0011$), and education ($\beta = −0.21$) were all significantly and negatively associated with forest dependency ($p < 0.001$ for all). Larger families, higher income, and better education were linked to reduced reliance on forest resources, likely due to diversified livelihoods and greater access to alternatives. These findings support previous research on the socio-economic drivers of forest use and conservation [105,106]. The inverse relationship between education and forest dependency ($\beta = −0.21$, $p < 0.001$), and its positive contribution to conservation behavior ($\beta = 0.31$, $p < 0.001$), the environmental awareness is enhanced and pressure on forest resources is reduced through education [105,106]. Conservation behavior was analyzed using a logistic regression model, with a binary variable indicating whether households participated in conservation. Education had a positive and significant effect ($\beta = 0.31$, $p < 0.001$), suggesting that higher education promotes environmental awareness. Income also showed a positive association ($\beta = 0.0017$, $p = 0.005$), implying that wealthier households are more likely to engage in sustainable practices. Similarly, household size was positively linked to conservation behavior ($\beta = 0.28$, $p = 0.002$), possibly due to greater labor availability for community forest management activities [Table 5].

**Table 4. Pearson correlation analysis: Population growth Vs Forest cover change, Yayo coffee forest biosphere reserve from 1984 to 2024, Southwest Ethiopia, Supplementary Table (S1 Table).**

| Factors | Population growth | Forest cover change | Year of change |
|---|---|---|---|
| Population growth | 1 | −0.998* | 0.999* |
| Sig. value | | 0.043 | 0.020 |

**Table 5. Socio-economic determinants of forest dependency and conservation behavior in Yayo coffee forest Biosphere Reserve, Southwest Ethiopia, 2024, Supplementary Table (S2 Table).**

| Variable | LR on forest resources | | LR conservation behavior | |
|---|---|---|---|---|
| | Coefficient (A) | *p-value* | Coefficient (B) | *p-value* |
| Family Size | −0.32 | 0.000 | 0.28 | 0.002 |
| Income Level (ETB) | −0.0011 | 0.000 | 0.0017 | 0.005 |
| Education status (Years) | −0.21 | 0.000 | 0.31 | 0.000 |

LR* - Linear Regression, LR- Logistic Regression.

Similarly, income's role in decreasing forest dependency ($\beta = -0.0011$, $p < 0.001$) and increasing conservation participation ($\beta = 0.0017$, $p = 0.005$) aligns with the finding that income diversification enables households to meet their demands without over-exploiting forests [107]. Although it appears contradictory to assume that family size would be positively associated with conservation conduct, this study found that larger families were less dependent on the forest ($\beta = -0.32$, $p < 0.001$) and more involved in conservation activities ($\beta = 0.28$, $p = 0.002$), consistent with those who argued that larger families are capable of rendering more effort and labor towards conservation [108].

## 4. Conclusion and policy implications

The Yayo Coffee Forest Biosphere Reserve is crucial for local livelihoods, despite the growing threat of increasing population, expansion of farmlands, and the weak coordination at the institutional level. Traditional systems of governance, such as Tuullaa and Jaarsa Biyya, work to regulate resource use, while formal institutions face the problem of limited capacity and community engagement. A strong negative correlation between population growth and forest cover results in an 11.6% loss over four decades, and this indicates pressure on natural resources. It requires the integration of indigenous governance into formal systems, environmental awareness, diversified income, and the enforcement of land-use policy that balances ecological protection with community needs. The study also suggests revision in land allocation policy by adopting a "Comprehensive-to-Detail" approach: First, by undertaking a detailed survey of the available land and creating flexible standards that allow efficient and effective use of the land; and second, by continual research on this approach and other complementary strategies to minimize land consumption.

## Supporting information

**S1 Fig. Satellite Images Results of Land Use Land Cover in the Yayo coffee forest Biosphere Reserve from 1984 to 2024, Southwest Ethiopia [Hurumu, Doreni and Yayo districts].**
(DOCX)

**S1 Table. Pearson Correlation analysis Data: Population growth Vs Forest cover change, Yayo coffee forest Biosphere Reserve from 1984 to 2024, Southwest Ethiopia.**
(DOCX)

**S2 Table. Socio-Economic Determinants Data: Forest Dependency and Conservation Behavior in Yayo coffee forest Biosphere Reserve, Southwest Ethiopia, 2024.**
(DOCX)

## Author contributions

**Data curation:** Fikru Mosisa Hunde.

**Formal analysis:** Tefera Jegora Kapula.

**Methodology:** Adanech Asfaw Benti.

**Resources:** Adanech Asfaw Benti.

**Software:** Tefera Jegora Kapula.

**Supervision:** Fikru Mosisa Hunde.

**Writing – original draft:** Fikru Mosisa Hunde, Tefera Jegora Kapula.

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
