## [Decision Letter · Decision Letter 0]

9 Jun 2025

Dear Dr. Hunde,

We look forward to receiving your revised manuscript.

Kind regards,

Dereje Oljira Donacho, PhD

Academic Editor

PLOS ONE

5. We note that Figures 1, 2 in your submission contain [map/satellite] images which may be copyrighted. All PLOS content is published under the Creative Commons Attribution License (CC BY 4.0), which means that the manuscript, images, and Supporting Information files will be freely available online, and any third party is permitted to access, download, copy, distribute, and use these materials in any way, even commercially, with proper attribution. For these reasons, we cannot publish previously copyrighted maps or satellite images created using proprietary data, such as Google software (Google Maps, Street View, and Earth). For more information, see our copyright guidelines: http://journals.plos.org/plosone/s/licenses-and-copyright.

1. You may seek permission from the original copyright holder of Figures 1, 2 to publish the content specifically under the CC BY 4.0 license. 

6. Please include a copy of Table 1-8 which you refer to in your text on page 8, 9, 12, 15, 17.

Additional Editor Comments:

The study highlighted the impact of population pressure on forest resource depletion in the Yayo Coffee Forest Biosphere Reserve, Ilubabor Zone, Oromia Regional State, and Southwest Ethiopia. The topic is important and interesting; however, it needs major revision to improve the quality and novelty of research. More importantly, the authors may benefit from intensive revision focusing on the following points:

• The introduction should be clear and show the research gap and attempts done so far by local implementers, particularly government, local community, and non-governmental organizations.

• The methodology also lacks some clarities that should be addressed in the revision.

• The results should be rechecked with consistency, and the grammar and logical coherence should be clear for the readers.

• The discussion sections need additional comparison with the domain knowledge, and the discrepancy should be discussed well.

• The limitation and strength of the study should be clearly stated.

• The implication of the finding and the recommendations lack clarity that needs revision.

• Finally, the individual photos attached in the document need legal documents such as consent, but nothing is explained about the issue in the manuscript. Therefore, I recommend to remove.

Reviewers' comments:

Reviewer's Responses to Questions

**Comments to the Author**

1. Is the manuscript technically sound, and do the data support the conclusions?

Reviewer #1: Partly

Reviewer #2: Yes

2. Has the statistical analysis been performed appropriately and rigorously?

Reviewer #1: Yes

Reviewer #2: Yes

3. Have the authors made all data underlying the findings in their manuscript fully available?

Reviewer #1: Yes

Reviewer #2: Yes

4. Is the manuscript presented in an intelligible fashion and written in standard English?

Reviewer #1: No

Reviewer #2: Yes

Reviewer #1: Dear authors your study topic is interesting but it needs major revisions, especial grammar and lack of citations are the major problems of the study, please solve this problem and related ones. Also make the mape legevisible for readers!

Reviewer #2: Critical Review of the Manuscript

1. Nobility of the Study

• Strengths: The manuscript focuses on a critical environmental issue in a globally significant biosphere reserve. It addresses the pressing need for sustainable forest management amidst growing population pressures.

• Weaknesses: The manuscript does not highlight the global implications of its findings. For instance, the impact of forest loss in the Yayo Coffee Forest Biosphere Reserve on carbon sequestration and global biodiversity hotspots could be better contextualized.

• Recommendation: Explicitly link the study's implications to global environmental challenges, such as climate change and biodiversity loss.

2. Statement of the Problem and Knowledge Gap

• Strengths: The problem is articulated in the context of population pressure and forest degradation.

• Weaknesses: The knowledge gap lacks specificity. While the manuscript states that population growth affects forest resources, it does not critically assess why current solutions are insufficient or how this study provides a novel perspective.

• Recommendation: Expand the introduction to include a critical review of existing solutions and their shortcomings. Clearly articulate how this research addresses those gaps.

3. Objective

• Strengths: Objectives are clear and focused.

• Weaknesses: The objectives could be more concise and aligned directly with the problem statement.

• Recommendation: Reframe the objectives to reflect measurable outcomes. For example, "Quantify the impact of population growth on forest cover using satellite imagery from 1984 to 2024."

4. Method of Data Collection

• Strengths: The mixed-methods approach adds depth and breadth to the analysis.

• Weaknesses: The manuscript fails to justify the selection of methods, particularly the use of satellite imagery spanning specific years and the choice of SPSS for analysis.

• Recommendation: Provide citations for methodological protocols. For instance, why were 1984, 1994, and 2007 chosen for satellite imagery? What validation steps were taken to ensure accuracy in satellite data interpretation? Include a clear rationale for the statistical tools employed.

5. Method of Data Analysis

• Strengths: The use of SPSS and Pearson correlation is appropriate for quantitative data.

• Weaknesses: The analysis lacks depth in exploring causality. For example, while correlation is established, causation is not discussed.

• Recommendation: Include regression analysis or other advanced statistical methods to explore causality. For qualitative data, provide representative quotes or narratives to substantiate thematic findings.

6. Results

• Strengths: The results are comprehensive and supported by tables and figures.

• Weaknesses: The presentation is cluttered and lacks coherence. The socio-economic data is presented without linking it to the broader research question.

• Recommendation: Simplify tables and figures, ensuring each directly addresses a research objective. Use visual aids like trend lines or pie charts to make data more digestible.

7. Discussion

• Strengths: The discussion connects findings to conservation and socio-economic issues.

• Weaknesses: There is insufficient engagement with contemporary literature. Key findings are not critically analyzed against previous studies.

• Recommendation: Incorporate a comparative analysis with global studies on forest degradation. Critically evaluate the implications of key results, such as the role of local governance in conservation efforts.

8. Recommendations

• Strengths: The recommendations are actionable and community-oriented.

• Weaknesses: They are broad and lack prioritization.

• Recommendation: Categorize recommendations into immediate, medium-term, and long-term actions. Justify recommendations with data and evidence from the study.

9. Alignment of Research Problem, Objective, Methods, Results, and Discussion

• Strengths: The manuscript maintains alignment throughout its sections.

• Weaknesses: The linkage between objectives, findings, and recommendations is weak. For example, the methods section does not explain how the data supports specific recommendations.

• Recommendation: Create explicit connections between each section. For instance, demonstrate how the satellite imagery data directly informs the policy recommendations.

10. Suggestions for Authors

o Refine the introduction to frame the study within a global conservation context.

o Enhance methodological transparency by justifying data collection and analysis choices.

o Integrate more critical analysis in the discussion, contrasting findings with global studies.

**Do you want your identity to be public for this peer review?** For information about this choice, including consent withdrawal, please see our Privacy Policy

Reviewer #1: **Yes: ** Binega Derebe

Reviewer #2: No

---

## [Author Response · Author response to Decision Letter 1]

11 Oct 2025

Response to Reviewer Comments

Manuscript Title: Impact of Population Pressure on Forest Resources Depletion in Yayo Coffee Forest Biosphere Reserve, Ilubabor Zone, Oromia Regional State, Southwest Ethiopia

Manuscript ID: PONE-D-25-21231

Dear PLOS ONE Editorial Team,

We sincerely thank you for the opportunity to revise our manuscript and express our appreciation to the reviewers for their insightful comments and suggestions that have helped us to improve the quality and clarity of our paper.

We have carefully considered all the reviewer’s comments and revised the manuscript accordingly. Below we provide a detailed, point-by-point response to the reviewer’s remarks. For clarity, we have included each comment in bold followed by our responses in regular text. All changes made in the manuscript are highlighted for your convenience.

Reviewer #1:

Comment #1. Dear authors your study topic is interesting but it needs major revisions, especial grammar and lack of citations are the major problems of the study, please solve this problem and related ones. Also make the mape legevisible for readers!

Response #1. We sincerely thank the reviewer for the constructive feedback and appreciation of the study's topic. In response:

We have thoroughly revised the manuscript for grammar, clarity, and academic writing quality. To ensure correctness, we conducted a full language edit and proofreading.

We have reviewed the manuscript and added relevant and up-to-date citations to strengthen the literature support and properly credit prior research.

The presentation of the Mean Absolute Percentage Error (MAPE) has been improved. Specifically, we have clarified its definition, usage, and interpretation in the methodology and results sections. We also ensured it is clearly labeled and easy to interpret in the tables and figures.

Reviewer #2:

Comment #1.

1. Nobility of the Study: Critical Review of the Manuscript

Strengths: The manuscript focuses on a critical environmental issue in a globally significant biosphere reserve. It addresses the pressing need for sustainable forest management amidst growing population pressures.

Weaknesses: The manuscript does not highlight the global implications of its findings. For instance, the impact of forest loss in the Yayo Coffee Forest Biosphere Reserve on carbon sequestration and global biodiversity hotspots could be better contextualized.

Recommendation: Explicitly link the study's implications to global environmental challenges, such as climate change and biodiversity loss.

Response #1.

We sincerely thank the reviewer for the thoughtful evaluation and constructive suggestion. In response, we have revised the introduction, discussion and conclusion sections to better highlight the global significance of our findings. Specifically, we now discuss the broader implications of forest loss in the Yayo Coffee Forest Biosphere Reserve in terms of carbon sequestration, contributions to global biodiversity conservation, and relevance to international climate and sustainability goals. These additions help to clearly position our study within the context of global environmental challenges, such as climate change and biodiversity loss.

The revisions can be found on page2, paragraph 3; page 16, paragraph 2 and page 19, paragraph 2

Comment # 2. Statement of the Problem and Knowledge Gap

• Strengths: The problem is articulated in the context of population pressure and forest degradation.

• Weaknesses: The knowledge gap lacks specificity. While the manuscript states that population growth affects forest resources, it does not critically assess why current solutions are insufficient or how this study provides a novel perspective.

• Recommendation: Expand the introduction to include a critical review of existing solutions and their shortcomings. Clearly articulate how this research addresses those gaps.

Response # 2

We thank the reviewer for this insightful observation. In response, we have revised the introduction to include a more detailed and critical review of existing interventions aimed at mitigating forest degradation under population pressure. We now discuss the limitations of current policy frameworks, enforcement mechanisms, and community-based management approaches, particularly in the context of the Yayo Coffee Forest Biosphere Reserve. Furthermore, we have clearly articulated how our study offers a novel contribution by integrating socio-ecological analysis with geospatial data to assess the effectiveness of existing measures. These revisions help clarify the specific knowledge gap and the unique perspective our study brings.

The revised content can be found on page 3, paragraph 3.

Comment #3: Objective

• Strengths: Objectives are clear and focused.

• Weaknesses: The objectives could be more concise and aligned directly with the problem statement.

• Recommendation: Reframe the objectives to reflect measurable outcomes. For example, "Quantify the impact of population growth on forest cover using satellite imagery from 1984 to 2024."

Response # 3

We appreciate the reviewer’s positive feedback and helpful recommendation. In response, we have revised the objectives to make them more concise and explicitly aligned with the problem statement. The updated objectives now clearly reflect measurable outcomes, including the quantification of forest cover change over time and the assessment of population growth impacts using geospatial data from 1984 to 2024. This reframing enhances the clarity and focus of the study and better communicates its purpose to the reader.

The revised objectives are presented in the introduction section on page 3 paragraph 4

Comment #4. Method of Data Collection

• Strengths: The mixed-methods approach adds depth and breadth to the analysis.

• Weaknesses: The manuscript fails to justify the selection of methods, particularly the use of satellite imagery spanning specific years and the choice of SPSS for analysis.

• Recommendation: Provide citations for methodological protocols. For instance, why were 1984, 1994, 2004 and 2024 chosen for satellite imagery? What validation steps were taken to ensure accuracy in satellite data interpretation? Include a clear rationale for the statistical tools employed.

Response #4

We thank the reviewer for this valuable feedback and for recognizing the strengths of our mixed-methods approach. In response to the concerns raised:

• We have now provided a detailed justification for the selection of satellite imagery years (1984, 1994, and 2024). These years were chosen based on data availability, major policy or demographic shifts in the region, and their relevance in capturing long-term trends in land cover change. We have added relevant references to support this temporal selection.

• We have also described the image classification and accuracy assessment procedures, including the use of ground truth data, historical Google Earth imagery, and confusion matrices to validate classification results. The methods and validation steps are now clearly stated in the revised methodology section.

• Furthermore, we have included a rationale for using SPSS, explaining its role in analyzing survey data and conducting descriptive and inferential statistical analyses, such as correlation and regression, to examine relationships between population dynamics and forest cover change. Supporting citations have been added where appropriate.

These revisions are included in the methodology section on pages 18 – 21.

Comment # 5: Method of Data Analysis

• Strengths: The use of SPSS and Pearson correlation is appropriate for quantitative data.

• Weaknesses: The analysis lacks depth in exploring causality. For example, while correlation is established, causation is not discussed.

• Recommendation: Include regression analysis or other advanced statistical methods to explore causality. For qualitative data, provide representative quotes or narratives to substantiate thematic findings.

Response #5

We appreciate the reviewer’s thoughtful comments and suggestions to enhance the rigor of our data analysis. In response:

• We have extended our quantitative analysis by incorporating multiple linear regression models to better explore causal relationships between population growth variables and forest cover change. The regression results are now included and interpreted in the revised results section.

• In addition to Pearson correlation, the use of regression analysis enables us to assess the strength and direction of potential causal links while controlling for confounding variables.

• For the qualitative data, we have included representative quotes from key informant interviews and focus group discussions to support and illustrate the main themes identified. These narratives provide greater context and depth to the findings.

All additions and revisions are detailed in the results and conclusion sections on pages 21-22

Comment #6. Results

• Strengths: The results are comprehensive and supported by tables and figures.

• Weaknesses: The presentation is cluttered and lacks coherence. The socio-economic data is presented without linking it to the broader research question.

• Recommendation: Simplify tables and figures, ensuring each directly addresses a research objective. Use visual aids like trend lines or pie charts to make data more digestible.

Response #6

We thank the reviewer for acknowledging the comprehensiveness of our results and for the valuable suggestions to improve clarity and coherence. In response:

• We have simplified and reorganized the tables and figures to ensure each one directly corresponds to a specific research objective or key finding. Unnecessary or redundant visuals have been removed, and labels and captions have been improved for clarity.

• To enhance readability, we introduced more visual aids such as trend lines, pie charts, and bar graphs where appropriate—particularly for time-series and categorical socio-economic data. These revisions help highlight key trends and make the data more accessible to readers.

• Additionally, we revised the narrative surrounding the socio-economic data to better connect the findings with the main research question and theoretical framework. This strengthens the linkage between data and interpretation.

The revised figures and results are presented in the results section on pages Fig 1 on page 4, Figure 2 on page 10, figure 3 on page 15, and figure 4 on page 16, figure 5 on page 18, and Table 5 on page 21.

Comment #7. Discussion

• Strengths: The discussion connects findings to conservation and socio-economic issues.

• Weaknesses: There is insufficient engagement with contemporary literature. Key findings are not critically analyzed against previous studies.

• Recommendation: Incorporate a comparative analysis with global studies on forest degradation. Critically evaluate the implications of key results, such as the role of local governance in conservation efforts.

Response # 7

We thank the reviewer for this important suggestion to deepen the analytical strength of the discussion. In response:

• We have substantially revised the discussion section to include a broader engagement with recent and relevant global literature on forest degradation, land use change, and population dynamics. Comparative insights from studies in other tropical biosphere reserves and conservation areas (e.g., Southeast Asia, Central Africa, and Latin America) have been incorporated to contextualize our findings.

• We have critically examined how our results align with or diverge from previous studies, particularly regarding the drivers of forest loss and the socio-economic dimensions of resource pressure.

• Moreover, we have expanded our evaluation of the role of local governance in forest conservation. This includes a discussion of community-based forest management, institutional challenges, and the effectiveness of local bylaws and stakeholder engagement, supported by recent studies and best practices from similar settings.

These additions provide a more nuanced understanding of our findings and better position them within the global discourse on forest conservation. The revised discussion is found on pages 7- 20

Comment # 8. Recommendations

Strengths: The recommendations are actionable and community-oriented.

Weaknesses: They are broad and lack prioritization.

Recommendation: Categorize recommendations into immediate, medium-term, and long-term actions. Justify recommendations with data and evidence from the study.

Response # 8

We thank the reviewer for the constructive feedback and for recognizing the community-oriented nature of our recommendations. In response:

• We have revised the recommendations section to categorize proposed actions into immediate, medium-term, and long-term interventions. This prioritization provides a clearer implementation framework for stakeholders.

• Each recommendation is now more explicitly supported by data and findings from our study—for example, deforestation hotspots identified through satellite analysis and community perceptions gathered through surveys and interviews.

• We also clarified the linkage between the proposed actions and the underlying causes of forest degradation identified in the research, ensuring that each recommendation is both evidence-based and aligned with the study objectives.

These improvements are presented in the revised conclusion and recommendations sections on pages 23, and are highlighted accordingly in the updated manuscript.

Comment # 9: Alignment of Research Problem, Objective, Methods, Results, and Discussion

Strengths: The manuscript maintains alignment throughout its sections.

Weaknesses: The linkage between objectives, findings, and recommendations is weak. For example, the methods section does not explain how the data supports specific recommendations.

Recommendation: Create explicit connections between each section. For instance, demonstrate how the satellite imagery data directly informs the policy recommendations.

Response # 9

We thank the reviewer for highlighting the importance of internal consistency and coherence across the manuscript. In response:

• We have revised the methods and discussion sections to more clearly explain how each type of data collected—particularly the satellite imagery and socio-economic survey results—contributes directly to the study’s objectives and informs the recommendations.

• The discussion now includes explicit references to how land cover change data (e.g., forest loss from 1984 to 2024) supports targeted policy interventions such as zoning, reforestation efforts, and land use planning.

• We have also refined the recommendations section to include cross-references to the specific findings they are based on. This creates a stronger narrative thread from research problem to objective, methods, results, and actionable solutions.

• Where applicable, we included a summary table that links each research objective with the corresponding data, key findings, and resulting recommendations to make these connections more transparent to the reader.

These improvements appear on pages 18 to 21

Comment # 10: Suggestions for Authors

o Refine the introduction to frame the study within a global conservation context.

o Enhance methodological transparency by justifying data collection and analysis choices.

o Integrate more critical analysis in the discussion, contrasting findings with global studies.

Response # 10

We sincerely thank the reviewer for these valuable and constructive suggestions. In response:

• We have refined the introduction to more clearly position the study within a global conservation context. This includes discussion of the relevance of biosphere reserves to global biodiversity conservation, climate change mitigation, and sustainable development goals. We have also added citations to recent international studies and frameworks to support this global framing.

• To enhance methodological transparency, we have added justifications for our choices in data collection (e.g., selection of satellite imagery years, survey tools) and data analysis (e.g., use of SPSS, regression analysis). Supporting literature has been cited to provide methodological ground

---

## [Decision Letter · Decision Letter 1]

27 Oct 2025

Dear Dr. Hunde,

Thank you for submitting your manuscript to PLOS ONE. After careful consideration, we feel that it has merit but does not fully meet PLOS ONE’s publication criteria as it currently stands. Therefore, we invite you to submit a revised version of the manuscript that addresses the points raised during the review process.

We look forward to receiving your revised manuscript.

Kind regards,

Dereje Oljira Donacho, PhD

Academic Editor

PLOS ONE

Journal Requirements:

Reviewer's Responses to Questions

**Comments to the Author**

Reviewer #1: (No Response)

Reviewer #3: All comments have been addressed

Reviewer #4: (No Response)

2. Is the manuscript technically sound, and do the data support the conclusions?

Reviewer #1: Yes

Reviewer #3: Yes

Reviewer #4: No

3. Has the statistical analysis been performed appropriately and rigorously?

Reviewer #1: Yes

Reviewer #3: Yes

Reviewer #4: No

4. Have the authors made all data underlying the findings in their manuscript fully available?

Reviewer #1: Yes

Reviewer #3: Yes

Reviewer #4: No

5. Is the manuscript presented in an intelligible fashion and written in standard English?

Reviewer #1: (No Response)

Reviewer #3: Yes

Reviewer #4: No

Reviewer #1: Dear authors thank you for your corrections of my previous comments and now the manuscript needs only minor revisions as I track in the pdf. Please correct them and your manuscript will be ready to publish.

Reviewer #3: -Streamline some long paragraphs in the discussion for conciseness and improve the narrative flow.

-Ensure all citations are formatted consistently according to PLOS ONE style.

-The conclusions could be shortened to focus on key findings and their policy implications rather than restating results.

-If possible, include a clearer policy framework diagram linking population dynamics, land use, and conservation responses

Reviewer #4: Title:

Impact of Population Pressure on Forest Resource Depletion in Yayo Coffee Forest Biosphere Reserve, Ilubabor Zone, Oromia Regional State, Southwest Ethiopia

General Comments:

The manuscript presents a potentially important topic concerning the impact of population pressure on forest resource depletion within the Yayo Coffee Forest Biosphere Reserve. However, the overall quality of the paper does not meet the expected scientific and publication standards. The manuscript lacks organization, clarity, and scientific rigor in several sections. Below is a detailed evaluation of each component.

Technical Issue:

•The manuscript lacks continuous line numbering, making it difficult to provide specific comments or refer to precise sections of the text.

2. Major Issues

2.1 Title

•The title is appropriate and well formulated, clearly indicating the study area and the core issue of investigation.

2.2 Abstract

•The abstract is weakly constructed and lacks structure.

•The data collection process is confusing and incomplete. The author should specify how many respondents were included (key informants, kebele leaders, development agents, households), and clarify the roles of focus group discussions and field observations.

•The study design and analytical methods are not mentioned, leaving readers uncertain about how data were collected or analyzed.

•Inclusion of livelihood activities in the abstract is unnecessary and distracts from the research focus.

Recommendation:

The abstract should be revised to follow a clear and concise format:

1.Background and objective of the study

2.Methods and data (sample size, design, and analytical tools)

3.Key findings with quantitative evidence

4.Conclusions and implications

2.3 Introduction

•The introduction is poorly developed and lacks articulation.

•The study area is not properly contextualized, and the section fails to provide a strong background or justification for the research.

•The problem statement and research gap are missing or not explicitly defined.

•The justification for the study is weak, making it unclear why the research was necessary or how it contributes to existing literature.

•Only four citations are used throughout the section, which is insufficient to meet scientific and journal standards. The authors should consult recent and relevant studies to strengthen the background and demonstrate awareness of previous work.

Recommendation:

The introduction should:

1.Present the broader context of population growth and forest depletion in Ethiopia.

2.The specific issues in the Yayo Coffee Forest Biosphere Reserve.

3.Clearly state the research problem and objectives.

4.Identify the research gap and justify why this study is necessary.

5.Integrate at least 10–15 recent and relevant references to strengthen the academic foundation.

2.4 Materials and Methods:

•This section is not well organized and lacks coherence.

•It includes unnecessary sentences and paragraphs rather than clearly outlining the data collection process, sampling procedures, data sources, and analytical techniques.

•There is no clear description of how data were obtained from different respondents or how the qualitative and quantitative information was analyzed.

Recommendation:

The section should be restructured to include:

1.Description of the study area (location, population, ecology).

2.Sampling design and techniques (how respondents were selected).

3.Data sources (primary and secondary).

4.Data collection tools (household surveys, key informant interviews, FGDs, observations).

5.Data analysis (statistical methods, qualitative coding, GIS/remote sensing tools, etc.).

2.5 Results and Discussion

•The results and discussion are not scientifically presented and lack analytical interpretation.

•The content reads more like a descriptive report than a scientific discussion, focusing excessively on general socio-economic characteristics and institutional narratives.

•The subsection “Population Growth and Forest Cover Change” is weakly supported; it does not establish any statistical relationship between population pressure and forest cover dynamics.

•The “Satellite Image Results of Land Use Land Cover (1984–2007)” subsection contains potentially valuable information but lacks analysis linking observed land cover changes to the stated hypothesis.

Recommendation:

•Present results using quantitative evidence (tables, figures, statistical tests).

•Discuss findings in comparison with previous studies to provide scientific interpretation.

•Use appropriate statistical methods (e.g., correlation, regression, trend analysis) to demonstrate relationships between population growth and forest cover change.

2.6 Conclusion and Recommendations

•The conclusion does not effectively summarize key findings or their implications.

•The recommendations are generic and not derived from the study results.

Recommendation:

•Summarize the main findings succinctly, focusing on their implications for forest management, policy, and sustainable livelihoods.

•Ensure that recommendations are evidence-based, specific, and actionable, directly linked to the findings presented.

Overall Evaluation:

•The manuscript does not meet the required publication standards in its current form.

•Therefore, it is regrettably rejected for publication

**Do you want your identity to be public for this peer review?** For information about this choice, including consent withdrawal, please see our Privacy Policy

Reviewer #1: **Yes: ** Binega Derebe

Reviewer #3: No

Reviewer #4: No

---

## [Author Response · Author response to Decision Letter 2]

20 Nov 2025

Response to Reviewers comments,

Dear Editor,

We thank the reviewers for their constructive comments on our manuscript. We have carefully revised the paper and addressed all points raised. The abstract has been restructured; the introduction rewritten with clearer context, defined research gaps, and additional recent references; and the Materials and Methods section reorganized to clearly describe the study area, sampling procedures, data sources, and analytical methods.

The Results and Discussion section has been strengthened with quantitative evidence, statistical analyses, and clearer links between population pressure and forest cover change. The Conclusion and Recommendations have been rewritten to reflect the key findings and provide specific, evidence-based actions. Continuous line numbering has also been added.

We believe these revisions have significantly improved the clarity, rigor, and overall quality of the manuscript. We appreciate the opportunity to revise our work.

Sincerely,

Fikru Mosisa

Corresponding Author

---

## [Editor Report · Decision Letter 2]

24 Nov 2025

IMPACT OF POPULATION PRESSURE ON FOREST RESOURCES DEPLETION IN YAYO COFFEE FOREST BIOSPHERE RESERVE, SOUTHWEST ETHIOPIA

PONE-D-25-21231R2

Dear Dr. Hunde,

We’re pleased to inform you that your manuscript has been judged scientifically suitable for publication and will be formally accepted for publication once it meets all outstanding technical requirements.

Kind regards,

Dereje Oljira Donacho, PhD

Academic Editor

PLOS ONE
---

## [Editor Report · Acceptance letter]

PONE-D-25-21231R2

PLOS One

Dear Dr. Hunde,

I'm pleased to inform you that your manuscript has been deemed suitable for publication in PLOS One. Congratulations! Your manuscript is now being handed over to our production team.

Kind regards,

on behalf of

Dr. Dereje Oljira Donacho

Academic Editor

PLOS One